# Lysin Motif (LysM) Proteins: Interlinking Manipulation of Plant Immunity and Fungi

**DOI:** 10.3390/ijms22063114

**Published:** 2021-03-18

**Authors:** Shu-Ping Hu, Jun-Jiao Li, Nikhilesh Dhar, Jun-Peng Li, Jie-Yin Chen, Wei Jian, Xiao-Feng Dai, Xing-Yong Yang

**Affiliations:** 1School of Life Sciences, Chongqing Normal University, Chongqing 401331, China; hsp8293@163.com (S.-P.H.); ljp0501@163.com (J.-P.L.); jianwei7956@163.com (W.J.); 2c/o State Key Laboratory for Biology of Plant Diseases and Insect Pests, Department of Plant Pathology, Institute of Plant Protection, Chinese Academy of Agricultural Sciences, Beijing 100193, China; protectivecolor@yeah.net (J.-J.L.); chenjieyin@caas.cn (J.-Y.C.); 3Department of Plant Pathology, University of California Davis, Salinas, CA 93905, USA; ndhar@ucdavis.edu

**Keywords:** lysin motif (LysM) protein, plant-fungus interactions, chitin, immunity manipulation

## Abstract

The proteins with lysin motif (LysM) are carbohydrate-binding protein modules that play a critical role in the host-pathogen interactions. The plant LysM proteins mostly function as pattern recognition receptors (PRRs) that sense chitin to induce the plant’s immunity. In contrast, fungal LysM blocks chitin sensing or signaling to inhibit chitin-induced host immunity. In this review, we provide historical perspectives on plant and fungal LysMs to demonstrate how these proteins are involved in the regulation of plant’s immune response by microbes. Plants employ LysM proteins to recognize fungal chitins that are then degraded by plant chitinases to induce immunity. In contrast, fungal pathogens recruit LysM proteins to protect their cell wall from hydrolysis by plant chitinase to prevent activation of chitin-induced immunity. Uncovering this coevolutionary arms race in which LysM plays a pivotal role in manipulating facilitates a greater understanding of the mechanisms governing plant-fungus interactions.

## 1. Introduction

Being immobile, plants face multiple challenges during their lifetime including those from pathogens. Plants do not possess a circulating immune system like animals, nor do they generate antibodies to fight pathogens [1]. Thus, unique strategies to defend themselves against pathogen invasions have been designed by plants over the course of their evolution. Plants have evolved immune receptors as a means to detect and subsequently deter pathogens [2]. Immune receptors in plants can be divided into two main types, pattern recognition receptors (PRRs) and nucleotide-binding leucine-rich repeat receptors (NLR) [3]. PRRs have been employed as the primary defense against the pathogens. In general, PRRs are localized on the cell membrane and belong to a family of receptor kinases that generally include an extracellular ligand-binding domain, a transmembrane domain, and an intracellular kinase domain [4]. The extracellular domains of PRRs can specifically recognize molecular-associated molecular patterns (MAMP), such as the bacterial lipopolysaccharide (LPS), flagellin, EF-Tu (elongation factor thermo unstable), peptidoglycans, fungal chitin, and β-glucans, and ergosterols activate intracellular kinase domains through transmembrane domains which in turn potentiate downstream immune signals [5,6,7]. To break through this first level of plant’s immune system barriers, pathogens secrete small molecules called effectors into plant cells. These effectors disrupt mitogen-activated protein kinase (MAPK) signaling and its downstream transmission by inhibiting the translation and activity of PRRs and their complex signaling. Besides this, it can also affect plant immunity by interfering with several cellular processes such as the production of ROS, vesicle transport, callus deposition, and cell wall strengthening [5,7].

The LysM domain includes proteins that directly play a role either in the PRRs regulating immunity in plants or as effectors in pathogenic fungi that suppress plant immunity [8]. The LysM domain is a general peptidoglycan-binding module, which has a βααβ secondary structure with two helices stacked on the same side of the anti-parallel β-sheet. Generally, the LysM module contains about 50 amino acids and binds to peptidoglycans, chitin, and their derivatives [9]. The LysM domain was initially discovered in the antimicrobial enzyme lysozyme of a bacteriophage [8], and a growing number of studies have found that LysM domain proteins are widely distributed in eukaryotes and prokaryotes [10,11]. As a functional domain that helps evade host immunity, the function of LysM domain proteins relies on the binding properties of the LysM domain to the fungal chitin polysaccharide, which is an insoluble complex carbohydrate in the fungal cell wall that generally is degraded by plant chitinases [12]. The cell wall of fungal pathogens is the first point of contact with the host plant, and the main component of the fungal cell wall is chitin [12]. For plants, fungal chitin acts as a MAMP, which induces plant immunity by binding with the cognate PRRs in plants. In contrast, to evade the host defense response, LysM domain-containing effectors are secreted by fungal pathogens to protect their cell wall. Chitin as an essential component of the fungal cell wall acts as an important molecule in the manipulation of host immunity mediated by LysM proteins. This intermolecular interaction underlines the outcomes of finely controlled host–pathogen processes, which have become the basis for improving disease resistance in agricultural crops [13]. Thus, in this review, we provide a comprehensive appraisal of the biological characteristics of LysM proteins from plants and fungi, the mechanisms by which manipulation of host immunity occurs, and their interactive roles between plants and fungi to facilitate an understanding of the mechanism of LysM-mediated immunity and its significance to agriculture.

## 2. LysM Mediates Chitin Signaling in Plants

The LysM exists in the form of LysM-containing receptor-like kinases (LYKs) or non-kinase LysM [14]. LYKs are exclusively present in plants and are involved in regulation of plant immunity [14]. Plant LysM proteins are highly diverse, with at least six different types unlike the non-kinase types, suggesting that LysM proteins occur in a range of complex structural domains [14,15]. However, a biological function has not been assigned to most non-kinase LysM genes.

### 2.1. LysM in Arabidopsis and Rice

Recognition of pathogen-associated molecular patterns (PAMPs) by PRRs is the first step in plant defense, PAMPs are conserved elicitor molecules such as fungal chitin and chitooligosaccharides, bacterial peptidoglycan, and the maturation and transportation of PRRs is an important biological process. Chitin elicitor receptor kinase 1 (CERK1) is a classical LysM member, which is found in rice and *Arabidopsis* [16,17]. It functions as a PRR on plant cell membranes to recognize some conservative molecular patterns derived from pathogenic fungi and induce the host’s immune response (Figure 1). CERK1, which contains three LysM domains, plays a key role in the perception of chitin oligosaccharide inducers and transduction of *N*-acetylglucosamine. It shows a higher affinity to chitin oligosaccharide with longer residues. Immune responses induced by fungal MAMP chitin include MAPK activation, reactive oxygen species (ROS) production, and expression of defense genes [16,18,19,20].

Rice chitin receptor OsCERK1 matures in the endoplasmic reticulum from where it is transported to the ectoplasmic reticulum through the Golgi apparatus. During this process, stress-induced protein (Hop/Sti1) and heat shock protein 90 (Hsp90) can assemble one or more plant-specific Rho-type GTPase OsRac1, a member of Rho family Guanosine Triphosphate ases (GTPases), which complexes in the endoplasmic reticulum, and subsequent transport of OsCERK1 from the endoplasmic reticulum to the exoplasmic reticulum regulates the maturation of OsCERK1. The transport process relies on the regulation of the small GTPase secretion-related protein and ras-related protein 1 (GTPase Sar1) [21]. As a small guanine nucleotide regulatory (G) protein, OsRac1 is involved in the OsCERK1-containing receptor complex that regulates immunity after fungal chitin is perceived by rice cells [21,22]. A receptor-like cytoplasmic kinase (RLCK) member, including the probable serine/threonine protein kinase-like 27 (PBL27), being a downstream component of CERK1, plays an important role in chitin sensing and downstream signaling, which finally contributes to the modulation of chitin-induced immunity in *Arabidopsis* [23]. After CERK1 chitin recognition, PBL27 is phosphorylated by CERK1. Phosphorylated PBL27 subsequently participates in MAPK activation [23]. Similarly, the receptor-like cytoplasmic kinase in rice OsRLCK185 can also be phosphorylated by chitin-activated OsCERK1 and elicit MAPK responses [24,25]. OsRLCK185 also phosphorylates cytoplasmic ferritin OsDRE2a, rice homologs of the yeast Dre2 protein, to modulate a chitin-induced ROS burst [26]. Other RLCK VII members, including OsRLCK57, OsRLCK107, OsRLCK118, and OsRLCK176, are also involved in several chitin signaling pathways [27,28,29]. In addition, several other downstream components involved in CERK1-mediated signaling have also been identified in *Arabidopsis*. Plant U-box protein 4 (PUB4) is a ubiquitin ligase that interacts with CERK1. It is a positive regulator of chitin signaling through its role in ROS production and callose deposition upon perception of chitin oligosaccharide [30]. *Arabidopsis* receptor-like cytoplasmic kinase VII members including RLCK VII-4, RLCK VII-5, RLCK VII-7, and RLCK VII-8 function downstream of the PRRs involved in flagellin and chitin signaling, contributing to the callose deposition and ROS generation. RLCK VII-4 was specifically shown to be important for chitin signaling through its role in activation of the MAPK cascade upon perception of chitin thus connecting a PRR to MAPK signaling [31]. Taken together, these results indicate that the CERK1–RLCK signaling module plays a conserved role in the production of ROS in *Arabidopsis* and rice during a chitin-triggered immune response [21,22,23,24,25,26,27,28,29,30,31].

Interestingly, the chitin receptor CERK1 was shown to bind directly to chitin, with all three extracellular LysM domains required for chitin binding, but, more significantly, without any other interacting proteins, thus providing evidence that CERK1 is a major chitin-binding protein in plants [32]. The conserved juxtamembrane (JM) paramembrane domain regulates the kinase activity of CERK1 in *Arabidopsis*. In addition, the JM paramembrane domain also plays a functionally conserved role in the activation of the chitin-sensing signal in *Arabidopsis* [33]. Finally, the dimerization of chitin-induced CERK1 is the key biological process that is necessary for immune signal transduction induced by chitin [34]. For instance, ligand-induced CERK1 homodimerization is required for molecular patterns of ligand perception, receptor activation and immune signaling transduction [34,35].

Protein phosphorylation is also a key regulatory step in plant LysM signaling [23]. First, as CERK1 is a major chitin-binding protein that is directly involved in chitin binding by LysM, its chitin signal perception and transmission are dependent on post-translational modifications and kinase activity [32]. It is known that CERK1 phosphorylates receptor-like cytoplasmic kinases PBL27 in *A. thaliana* [23,36]. Similarly, in rice, the guanine nucleotide exchange factor for OsRac1 OsRacGEF1 is phosphorylated by the cytoplasmic kinase OsCERK1 [22]. CERK1 also interacts with LYK5 to sense chitin oligosaccharide and activate the downstream components of the chitin signaling cascade; subsequently, LYK5 separates from CERK1 to enter vesicles and undergoes endocytosis by phosphorylation with CERK1 [37]. After chitin sensing, CERK1 recruits the CERK1-interacting protein phosphatase 1 (CIPP1), a predicted Ser/Thr phosphatase, to dephosphorylate Tyr^428^ and dampen CERK1 signaling [38].

Other RLK and RLP containing the LysM domain also exist in rice and *Arabidopsis* and are involved in chitin signaling. For instance, chitin exciton-binding proteins (CEBiPs) were the first discovered plant LysM RLPs in rice [39]. Analysis of the molecular structure of the OsCEBiP chitin recognition domain showed that the crystal structures of a free ectodomain (ED) of OsCEBiP (OsCEBiP-ECD) contains three tandem LysMs, followed by a new structural fold of cysteine-rich domains [40]. Structural analysis also showed that chitin binding cannot significantly affect the conformation of OsCEBiP, but chitin in a “sliding mode” causes OsCEBiP aggregation, which plays a key role in the perception and transduction of several chitin inducers in rice cells [39,40,41]. Furthermore, OsLYP4 and OsLYP6, as LysM-RLPs, also show chitin affinity, possibly as small chitin receptors [42], both of which can form homo- and heterodimers and interact with CEBiP to participate in signal transduction of peptidoglycan (PGN) and chitin [42,43]. OsCERK1 can also be used as a link to bind with OsLYP4 and OsLYP6 [44]. While the OsCEBiP homolog in *Arabidopsis*, lysin motif domain-containing glycosylphosphatidylinosi-tol-anchoredprotein2 (LYM2), shows similarity to the rice CEBiP, it is not involved in signaling despite the high affinity for binding with the *p*-chitin oligosaccharide [44]. LYM2 mediates a reduction in molecular flux via plasmodesmata in the presence of chitin oligosaccharide to induce the defense responses against the fungal pathogen *Botrytis ahlial*, but LYM2 is not required for CERK1-mediated chitin-triggered defense responses, indicating that there are at least two independent chitin-activated response pathways in *Arabidopsis* [44]. In addition, several other CEBiP homologs have been found in *Arabidopsis* to be involved in the regulation of chitin signal transduction and plant immunity, including LysM RLK1-interacting kinase 1 (LIK1), LysM receptor kinase 3 (LYK3), LysM receptor kinase 4 (LYK4), and LysM receptor kinase 5 (LYK5) [37,44,45,46,47]. Several chitin oligomers can induce heterodimerization of AtLYK5 and homodimerization of AtCERK1, but the chitin-induced AtLYK5–AtCERK1 heterodimer is stronger than the AtCERK1 homodimer because only AtCERK1 contains a kinase domain, which has also been shown to be responsible for initiating intracellular chitin signaling [48].

Negative regulatory factors also play a key role in the expression of basic and early pathogen-induced defense genes and protection of plant growth from sustained chitin-induced immune damage [36]. For instance, AtLYK3 and LIK1 negatively regulate chitin-induced plant defense and act as components of the cytoplasmic kinase PBL27 complex. Plant U-box protein 12 (PUB12) negatively regulates chitin-induced immunity [36,45,47]. Chitin-activated AtCERK1 dephosphorylates its 428 tyrosine residue by recruiting CIPP1, which can be used as a negative feedback mechanism to block chitin signal transduction [38]. However, how plants balance these positive and negative signaling regimes or whether they rely on other signaling molecules is still unknown.

Finally, immune response is a complex and precise network that includes multiple signaling cross-talks during plant-pathogen interactions, and LysMs are no exception [46]. AtLYK3 is involved in the hormone abscisic acid (ABA)-mediated negative cross-talk with plant’s immune responses [46]. LIK1 positively regulates jasmonic acid/ethylene (JA/ET) hormone signaling and negatively regulates chitin-induced plant defense responses [47]. The LysM domain of EMBRYO SAC 1 (OsEMSA1A) is responsible for the development and function of embryo blastocysts [49]. CERK1 is required in response to both arbuscular mycorrhizal (AM) symbiosis and plant salt stress [50,51,52,53]. These results suggest that CERK1, in addition to its involvement in the immune responses as RLKs, also performs two or more mechanistically independent functions. For instance, CERK1 may play a role in cell death control independent of chitin and is not dependent on its kinase activity [54]. The chitin oligosaccharide β-1,4-*N*-acetylglucosamine oligomers (GlcNAc), which serve as the only fungal cell wall glycosidic structure, also function as MAMPs in plants, but the unbranched β-1,3-glucan of cucumber polyphyte oligosaccharides can also cause CERK1 receptor-mediated PAMP-triggered immunity (PTI) in *Arabidopsis* [55]. *Arabidopsis* CERK1 mediates plant immunity in response to non-host resistance to *Fusarium oxysporum* [56]. OsCERK1 has recently been suggested to have an additional role in causing a bacterial lipopolysaccharide (LPS)-induced immune response in rice [54]. *Arabidopsis cerk1–4* mutant allele failed to accumulate extracellular protein hydrolysates, leading to enhancement of susceptibility or induction of senescence [54]. Since *cerk1–4* plants exhibited normal chitin responses, cell death was not related to the kinase activity of AtCERK1 [54]. The role of AtCERK1 in salt tolerance in *Arabidopsis* has also been reported, AtCERK1 interacts with calcium channel protein ANNEXIN 1 (ANN1), which is responsible for salt-induced calcium inward flow [50].

### 2.2. LysM in Other Plants

CERK1 homologs also have important biological functions in other plants, especially in immunity manipulations [57,58,59]. LysM-containing proteins are also differentiated functionally in a variety of cellular processes. LysM-containing protein 1 in mulberry (MmLYP1), a CEBiP homolog, has an inhibitory effect on insoluble chitin and relies on LysM-containing proteins 2 (MmLYK2, CERK1 homolog) during infections by ascomycetes [60]. In cotton, two LysM receptor-like kinases, Gh-LYK1 and Gh-LYK2, have a certain effect on the resistance of cotton to *Verticillium* wilt. All three LysM domains of Gh-LYK1 and Gh-LYK2 are required for their chitin-binding ability, and virus-induced gene silencing (VIGS) of Gh-LYK1 and Gh-LYK2 in cotton plants compromises resistance to *Verticillium dahliae*. However, Gh-LYK2 and Gh-LYK1 may contribute differentially to defense in cotton. Gh-LYK2, but not Gh-LYK1, is a pseudo-kinase, and the ED of Gh-LYK2 can induce ROS burst in plants [58]. GhWAK7A in cotton directly interacts with GhLYK5 and GhCERK1 and promotes chitin-induced GhLYK5–GhCERK1 dimerization, which plays an essential role in the chitin-induced response [61]. In addition, LysM-type receptor-like protein 1 (LYP1), LysM-containing receptor-like kinase 7 (LYK7), and extracellular LysM protein 3 (LysMe3) are membrane receptors that recognize chitin signals, which can activate downstream defense processes and enhance resistance to *V. dahliae* [62]. Unlike rice OsCERK1, LysM proteins in tomato present significant functional divergence [63]: in tomato, SlLYK10 and SlLYK12 participate in the regulation of AM symbiosis, SlLYK1 participates in the chitin-induced immune response, and SlLYK13 is involved in cell death [57,64]. The AtCERK1 homolog from apple, MdCERK1, improves resistance to *Alternaria alternata* [59]. *Pteris ryukyuensis* chitinase-A (PrChiA) has a chitin-binding and antifungal activity [65], while *Equisetum arvense* chitinase-A (EaChiA) does not [66]. The main structural difference between PrChiA and EaChiA is the number of LysM domains [66], but there are no data to suggest that this is related to differences in antifungal activity. In addition, the presence of LysM proteins in apples, grapes, potatoes, and apple trees have also been identified and proven to participate in chitin-induced immune responses [59,67,68,69]. LysM proteins in pea, petunia, and alfalfa are involved in the plant AM symbiosis [70,71,72,73].

## 3. LysM Inhibits Chitin Signaling in Fungi

Many fungal pathogens encode exosomal effectors containing the same LysM domain as plant’s chitin receptors. Most fungal pathogens inhibit chitin-induced plant immunity by blocking chitin sensing or signaling [74,75,76,77,78,79]. Ecp6 in *Cladosporium flavum*, the first LysM domain effector to be discovered, chelates chitin during an infection [74]. Chitin oligosaccharides released from the cell wall of the invading mycelium by cell lysis or degraded by plant chitinases bind to chitin oligomers to prevent plant perception, blocking the plant immunity triggered by chitin [74,75]. Subsequently, a batch of fungal LysM effectors involved in the inhibition of plant immunity have been discovered.

The *Colletotrichum higginsianum* extracellular LysM structural proteins ChELP1 and ChELP2 of the ascomycete fungus causing anthracnose disease in plants plays a dual role in attachment, cell function, and chitin-triggered plant immunosuppression. ChELP1 and ChELP2 play an important role in maintaining the virulence of fungi and their ability to penetrate *Arabidopsis* epidermis cells and cellophane membrane [79]. Fungalysin metalloprotease in the maize anthracnose fungus *Colletotrichum graminicola* (Cgfl) contains not only two LysM domains, but also zinc metalloproteinases of the conserved catalytic site HEXXH domain, and is widely conserved in fungi. Upon infestation of maize, Cgfl is specifically expressed during the biotrophic phase, and it was shown that the fungus uses both Cgfl-mediated and LysM protein-mediated strategies to control chitin signals by suppressing the chitinase activity of the host [80]. Similarly, the rice blast fungus *Magnaporthe oryzae* not only relies on the LysM protein SLP1, but also on the auxiliary activity family 9 protein (Aa9) homolog MoAa91 (of *M. oryzae*) which is required for surface recognition as well as for suppressing chitin-induced plant’s immune responses. MoAa91 is regulated by G protein signaling (RGS) and RGS-like proteins which are required for appressorium development and is also subjected to negative regulation by the transcription factor MoMsn2. The *MoAa91* mutant formed immature adherent cells at the artificially induced interface and the gene deletion mutants significantly reduced pathogenicity. Further studies showed that MoAa91 is secreted into the exosomal space where it binds with the chitin and chitin oligosaccharides by competing with the rice immune receptor CEBiP, resulting in the inhibition of chitin-induced plant’s immune responses [81]. ZtGT2 protein from the wheat pathogen *Zymoseptoria tritici* has the function of maintaining the outermost surface of the fungal cell wall helping in the extension of the hyphae on the solid surfaces. It is widespread in fungi and its loss resulted in the loss of pathogen virulence on the host surface arising from the handicap in hyphal extension. A mutation in the ZtGT2 orthologue in the pathogenic fungus *Fusarium graminearum* resulted in constitutive enhanced expression of several transmembrane and secreted proteins, including the important LysM domain Zt3LysM, a LysM domain chitin-binding virulence effector. Though the adhesion to leaf surfaces was unaffected in the cognate *F. graminearum* mutants, a severe impairment in hyphal growth in the mutants resulted in a similar loss of pathogenicity in this taxonomically unrelated fungus on the wheat ears [82].

*Mycosphaerella graminicola* causing the *Septoria tritici* blotch disease of wheat carries the effectors Mg1LysM and Mg3LysM that are homologs of the effector protein extracellular protein 6 (Ecp6) from the biotrophic leaf mold fungal pathogen *Cladosporium fulvum*. Previous work has shown that these fungal effectors can bind chitin, thus, in turn, protect fungal hyphae against plant-derived chitinases, likely by helping the pathogen to evade the host’s immune response activated through chitin receptors CERK1 and CEBiP [76,77]. The LysM structural protein of *Magnaporthe grisea* secreting an effector protein, secreted LysM protein 1 (SLP1), and *Rhizoctonia solani* LysM effector (RsLysM) inhibit chitin-triggered immunity by binding chitin, but they cannot protect the hyphae from hydrolysis [76,77,83]. A study of 18 putative LysM proteins in *Penicillium* spp. found that the expression levels of PeLysM1, PeLysM2, PeLysM3, and PeLysM4 were significantly increased during pathogen infection, but did not affect the fungal virulence. The PeLysM3 was also proved to regulate spore germination and growth rate [84].

However, fungal LysMs are not always only involved in manipulation of plant immunity resulting in deleterious effect on the host. A secreted LysM effector identified from the model arbuscular mycorrhiza (AM) fungal species *Rhizophagus irregularis* (RiSLM) is one of the highest expressed effector proteins in the intraradical mycelium during symbiosis with the host *Medicago truncatula*. In vitro binding experiments have shown that RiSLM binds to chitin oligosaccharides to protect fungal cell walls from chitinase. In addition, RiSLM can effectively interfere with the immune response triggered by chitin to subvert chitin-triggered immunity during symbiosis, which points to a common role of LysM effectors in both symbiotic and pathogenic fungi at some point in evolution [85,86]. On a similar note, the beneficial fungus *Trichoderma viride* employs effectors like Tal6 that sequesters GlcNAc oligomers, thereby interfering with fungal-derived *N*-acetylglucosamine perception by the plant’s surveillance machinery leading to protection of fungal hyphae from the host’s chitinases [87].

*N*-glycosylation as a post-transcriptional modification is a typical mechanism for effectors to be used extensively by fungal pathogens to modulate their ability to evade host immunity [88]. The Alg3-mediated *N*-glycosylation of Slp1 is required for its protein stability and chitin affinity. *N*-glycosylation with the aid of Slp1 is required to evade the host’s innate immunity [88]. ChELP1 and ChELP2 also are *N*-glycosylated [79]. These studies suggest that *N*-glycosylation is essential and is a common modification for fungal LysM effectors to achieve ultra-high chitin affinity [88]. Large-scale identification and comparison of *N*-glycosylation proteins at various biological development stages of the pathogenic fungus *Magnaporthe oryzae* showed that 559 *N*-glycosylation sites of 355 proteins were identified; most of them are involved in different biological development processes, such as mycelia, conidia, and cell attachment and differentiation [89].

## 4. LysM Interlocks Manipulation of the Immunity Responses between Plants and Pathogens

The current view of the inducible defense responses in plant-pathogen interactions is well captured in the so called zigzag model. In this model, the first induced defense is activated by the PRRs, and PRRs are cell surface receptors that recognize MAMP [90]. This defense response includes local cell wall strengthening, production of reactive oxygen species, and production and release of antimicrobial compounds, which together prevent most microbial invasions [1,3,7,90]. When pathogens successfully overcome the host’s defenses by using secreted effectors, effector-triggered susceptibility is established, which is a key element of the zigzag model [1]. LysM proteins, the important members that are present both in plants and fungi, should play an important role to interconnect the immunity modulation between plants and pathogens.

During the fungus-plant interactions, when fungi break through the first plant’s barrier and engage the host’s surface, plant LysM proteins work as the PRRs to recognize chitin and induce immunity. The plant employs the secretory chitinase to hydrolyze the fungal cell wall to release free oligochitin. Subsequently, defense signals are transmitted by the downstream LysM cytoplasmic kinase, triggering the plant’s immune response to suppress the proliferation of fungi. Conversely, fungi secrete LysM proteins to suppress the chitin-induced immunity. Microbial LysM proteins protect the fungal cell wall from hydrolysis by binding to the outermost surface or compete with plant LysM receptors to bind chitin fragments and ultimately surmount plant immunity to facilitate the fungal infection.

Thus, LysM proteins interconnect the immunity manipulation between plants and pathogens. This has been well-demonstrated in *Verticillium* spp. In cotton, several LysM proteins have a certain effect on the resistance of cotton to *Verticillium* wilt, including Gh-LYK1, Gh-LYK2, and GhLYK5 [58], through their chitin-binding ability. In *V. dahliae*, the effect of LysM of *Vd*2 contributes to the virulence of the pathogen and binds chitin to suppress chitin-induced immune responses, thereby protecting fungal hyphae against hydrolysis by the plant’s hydrolytic enzymes [91]. Pathogens not only use LysM effectors to block plant immunity by interacting with the chitin polymer to protect the mycelium, as *Verticillium aflalfae* also secretes the *Verticillium nonalfalfae* chitin-binding protein (VnaChtBP) containing a carbohydrate-binding motif 18 (CBM18) domain that interacts with the chitin polymer [92]. In addition, a recent study on *V. dahliae* found that the deletion of a polysaccharide deacetylase gene (VdPDA1) seriously affected the pathogenicity of *V. dahliae* [93]. VdPDA1 has a strong chitin oligomer deacetylase activity on soluble chitin oligosaccharides in the alkaline environment. Further studies revealed that deacetylated chitin oligosaccharides impede recognition of LysM receptors, leading to the blocking of chitin-sense immune signals transmission in the intracellular, which suppresses the host’s immune response during host-*V. dahliae* interaction [93]. Therefore, pathogenic fungi can block chitin signaling by (1) secreting extracellular body effectors with high affinity for chitin oligomers, (2) secreting effectors that compete with chitin to bind with plant LysM receptor kinases, (3) deacetylating chitin to avoid the immune response caused by chitin perception, (4) and mimicking chitin to bind plant LysM receptor kinases.

## 5. Conclusions and Perspectives for Future Studies

The coevolutionary arms race between fungi and plants results in a plants’ defense response to fungal chitin via the immunity manipulation between plants and pathogens. Successful fungi have evolved LysM effector proteins that interfere with plant LysM-mediated immune responses that eventually lead to plant susceptibility. In turn, plants have evolved LysM receptors to stimulate defense responses. When fungi evolve to produce new LysM effectors or interfere with host immunity in other ways, it is believed that plants will also respond with new LysM receptors to maintain plant immunity.

Far from understanding the LysM proteins in plants and fungi, priority issues to drive research in the near future include the following: (1) the current understanding of the fungal LysM effectors and their inhibitory mechanisms in plant chitin signaling needs further advancement, including the emerging role of other fungal effectors involved in interconnecting immunity modulation by both the host and the pathogen, or the discovery of unknown plant LysM receptors, etc.; (2) identification of the different/common characteristics of LysM proteins of plants and fungi and their evolutionary relationships (this would facilitate an understanding of the coevolution of how LysM interconnects the immunity manipulation between plants and pathogens); (3) except for the immunity manipulation function during plant-fungus interactions, what the other functions of LysM proteins are, especially in fungi (in addition to the members involved in immunity manipulation, the biological functions of some other members of LysM proteins are still unclear); and (4) fungal LysM proteins act as effectors to protect the cell wall from hydrolysis by plant chitinases and manipulate the immunity during plant-fungus interactions. However, the role of LysM proteins in AM symbiosis also needs further investigation to facilitate the use of beneficial microbes for plant health and to attain food security in a sustainable manner.

## Figures and Tables

**Figure 1 ijms-22-03114-f001:**
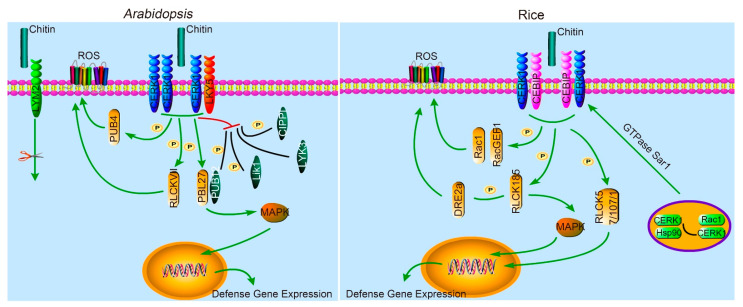
A model for fungal chitin-triggered immune signals involving LysM domain proteins in plants. LYM2: lysin motif domain-containing glycosylphosphatidylinositol-anchored protein 2; ROS: reactive oxygen species; CERK1: chitin elicitor receptor kinase 1; LYK5: LysM receptor kinase 5; PUB4: plant U-box protein 4; RLCK VII: receptor-like cytoplasmic kinase VII; PBL27: probable serine/threonine protein kinase-like 27; PUB12: plant U-box protein 12; Lik1: LysM RLK1-interacting kinase 1; LYK3: LysM receptor kinase 3; CIPP1: CERK1-interacting protein phosphatase 1; MAPK: mitogen-activated protein kinase; P: phosphorylation; CEBiP: chitin exciton-binding protein; Rac1: a member of Rho family (GTPases); RacGEF1: guanine nucleotide exchange factor for OsRac1; DRE2a: rice homologs of the yeast Dre2 protein; RLCK185: *Oryza sativa* receptor-like cytoplasmic kinase 185; RLCK57/107/118/76: receptor-like cytoplasmic kinase 57/107/118/76; GTPase Sar1: GTPase secretion-associated and ras-related protein 1; Hsp90: heat shock protein 90.

## Data Availability

Not applicable.

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
