# Peer review of "Lysin Motif (LysM) Proteins: Interlinking Manipulation of Plant Immunity and Fungi"

_ijms, 2021, doi:10.3390/ijms22063114_

Round 1
Reviewer 1 Report
The authors provided a historical review on plant and fungal Lysin Motif (LysM) proteins to illustrate how LysM proteins from plants and fungi are involved in the co-evolutionary manipulation of plant immunity by pathogens. This review article is written in very clearly but its language should be improved. There are many grammar issues and typos in the manuscript. Below are some examples. Hope the authors could carefully proofread and correct the English issues in their resubmission.
- Line 14, “the manipulation of plant immunity manipulation by pathogens”. One manipulation should be removed.
- Line 23, “including from” should be “including those from”
- Line 73, it cannot be inferred why “receptor-like kinases” are abbreviated as LYK.
- Line 90, is “Chint” in Figure 1 a typo? Should be “Chitin”?
- Line 168, “LYK2” has not been defined. Do you mean LYM2?
- Line 221, “Verticillium wilt” should be in italic.
- Line 344, “through interconnects” should be “through interconnecting”.
- Lines 352-353, “is still need further development” should be “needs further development”.
- Line 353, “other fungal effectors involves” should be “other fungal effectors involved in”.
- Line 358, “involves in” should be “involved in”.
Author Response
Dear Reviewer,
Thank you very much for your replies and assistance. We appreciate your comments and your kind suggestions for our manuscript (ijms-1127364) entitled “Lysin Motif (LysM) Proteins: Interlinking Manipulation of Plant Immunity and Fungi ”. We provide this letter to explain, point by point, the details of our revisions in the manuscript and our responses to the your comments as follows.
According to your comments, we have revised our manuscript. In addition, we have carefully checked through the whole manuscript and corrected some grammar mistakes. To make the changes easily viewable for you, we provided two revision (one including track changes, another without track changes). We hope the revised manuscript will satisfy your requests.
We are looking forward to hearing from you.
Kind Regards,
Xiao-Feng Dai and Xing-Yong Yang
Response to your comments:
The authors provided a historical review on plant and fungal Lysin Motif (LysM) proteins to illustrate how LysM proteins from plants and fungi are involved in the co-evolutionary manipulation of plant immunity by pathogens. This review article is written in very clearly but its language should be improved. There are many grammar issues and typos in the manuscript. Below are some examples. Hope the authors could carefully proofread and correct the English issues in their resubmission.
Reply:
Thanks for the suggestions,I have carefully revised the language of the entire manuscript and corrected the English issues.
Question1:Line 14, “the manipulation of plant immunity manipulation by pathogens”. One manipulation should be removed.
Question2:Line 23, “including from” should be “including those from”
Question3:Line 73, it cannot be inferred why “receptor-like kinases” are abbreviated as LYK.
Question4:Line 90, is “Chint” in Figure 1 a typo? Should be “Chitin”?
Question5:Line 168, “LYK2” has not been defined. Do you mean LYM2?
Question6:Line 221, “Verticillium wilt” should be in italic.
Question7:Line 344, “through interconnects” should be “through interconnecting”.
Question8:Lines 352-353, “is still need further development” should be “needs further development”.
Question9:Line 353, “other fungal effectors involves” should be “other fungal effectors involved in”.
Question10:Line 358, “involves in” should be “involved in”.
Reply:
Many thanks! We have corrected all the errors and also invited the native speaker, the outstanding Phytopatholigist Prof. Krishna Subbarao and Dr. Nikhilesh Dhar from UC-Daivs, to edit and polish the language according your suggestions.

Reviewer 2 Report
- Stating that plants do not have an immune system, is very misleading in this reviewer's opinion.
-Many inconsistencies are present in the text, this reviewer would suggest thorough corrections for terms and abbreviations.
Author Response
Dear Reviewer,
Thank you very much for your replies and assistance. We appreciate your comments and your kind suggestions for our manuscript (ijms-1127364) entitled “Lysin Motif (LysM) Proteins: Interlinking Manipulation of Plant Immunity and Fungi ”. We provide this letter to explain, point by point, the details of our revisions in the manuscript and our responses to the your comments as follows.
According to your comments, we have revised our manuscript. In addition, we have carefully checked through the whole manuscript and corrected some grammar mistakes. To make the changes easily viewable for you, we provided two revision (one including track changes, another without track changes). We hope the revised manuscript will satisfy your requests.
We are looking forward to hearing from you.
Kind Regards,
Xiao-Feng Dai and Xing-Yong Yang
Response to your comments:
Question1:Stating that plants do not have an immune system, is very misleading in this reviewer's opinion.
Reply:
Thanks for the suggestions,I have revised this sentence.
Question2:Many inconsistencies are present in the text, this reviewer would suggest thorough corrections for terms and abbreviations.
Reply:
The terms and abbreviations have been completely corrected as required,thanks for your reminding.

Round 2
Reviewer 2 Report
Dear authors,
This reviewer appreciates the extensive corrections and clarifications of the revised version. The text is much more fluid and statements are clear.
Thank you for taking the time to improve your very interesting review.
Best regards,
Author Response
Dear Reviewer,
During the final check through your manuscript, I realized a couple of minor editorial issues, that I would kindly ask you to amend prior to final acceptance.
line 16: The term straddle seems awkward to me.
Corrected it.
lines 39-44: This sentence is probably supposed to be divided into two.
Corrected it.
Lines 46-48: There are many more effector targets than just MAPK. Please correct the statement accordingly.
Corrected it.
Line 59: As far as I know, lysozyme was first described in eukaryotes. Please double check the facts.
Corrected it.
Lines 64-74: As you describe later, some LysM-containing proteins bind to chitooligosaccharides, and not just chitin. Please specify the statements in this paragraph and throughout the manuscript, where appropriate.
Corrected it.
Figure 1: In the top panel, chitin is misspelled. Please correct. Furthermore, the labels in the schemes are very hard to read. Please increase the font sizes accordingly.
Corrected it. Thank you!
Line 292: Colletotrichum higginsianum is an ASCOMYCETE fungus. Please amend.
Corrected it.
Line 366: How can LysM interlink the IMMUNITY between plant and pathogen? Immunity is an exclusive feature of the plant host. Please improve the chapter title.
Corrected it.
Line 411 should probably read plants
Corrected it.
Line 414: … immune responses that eventually lead to plant susceptibility.
Corrected it.
Line 412: ‘its’ should be replaced by ‘their’.
Corrected it.